# Pleistocene Aquatic Refuges Support the East–West Separation of the Neotropical Catfish Trichomycterinae (Siluriformes: Trichomycteridae) and High Diversity in the Magdalena, Guiana, and Paraná-Paraguay Basins

**Thais de Assis Volpi** [1] , **Marina Monjardim** [1] , **Luisa M. Sarmento-Soares** [2] **and Valéria Fagundes** [1,*]

1   Laboratório de Genética Animal, Departamento de Ciências Biológicas, Universidade Federal do Espírito Santo (LGA-CCHN-UFES), Av. Fernando Ferrari, 514, Goiabeiras, Vitória 29.075-910, Espírito Santo, Brazil; thaisvolpi@gmail.com (T.d.A.V.); marina.monjardim@gmail.com (M.M.)
2   Instituto Nacional da Mata Atlântica (INMA), Av. José Ruschi, 4, Centro, Santa Teresa 29.650-000, Espírito Santo, Brazil; sarmento.soares@gmail.com
*   Correspondence: valeria.fagundes@ufes.br

**Abstract:** (1) Background: Trichomycterinae represent 60% of the species in the family and, while seven genera comprise 1–3 species each, *Trichomycterus* and *Cambeva* have over 180 known species between them. Although integrative studies aimed to clarify the relationships within the subfamily, the diversity of species of *Trichomycterus* remains an open question. Herein, we explored an unprecedented sample to investigate the divergence in the lineages of *Trichomycterus*. (2) Methods: we recovered the phylogenetic relationships of the subfamily using 566 sequences (999 bp) of the mitochondrial gene cytochrome b, calculated intra- and intergroup distance percentages, and estimated divergence times. (3) Results: we recovered 13 highly supported and geographically structured lineages; intergenus divergence was 11–20%, while interspecies divergence was 3–11%; *Trichomycterus*, *Cambeva*, *Scleronema*, *Hatcheria*, *Eremophilus*, and *Ituglanis* were recovered as monophyletic, with three other highly divergent clades: Guiana Shield, Magdalena basin, and Tapajós basin. (4) Conclusions: We propose that the trans-Andean austral clades be allocated into *Hatcheria*, and the Guiana clade supports a new genus. We also observed that the headwaters nearest the Magdalena and Orinoco basins showed a high diversity and endemism of Trichomycterinae lineages. We discussed the role of geomorphological events and the climatic features which may explain cladogenesis events in Trichomycterinae.

**Keywords:** catfish; cladogenesis; genetic structure; geographic isolation; vicariance





## 1. Introduction

The Trichomycteridae are an assemblage of small-sized Neotropical freshwater catfishes with an adult size of less than 150 mm, inhabiting a diverse variety of habitats. The group is divided into eight subfamilies, of which the Trichomycterinae is the most diverse, with 60% of all taxa in the family [1]. So far, nine genera are recognized in Trichomycterinae: *Bullockia* Arratia, Chang, Menu-Marque and Rojas, 1978; *Eremophilus* Humboldt, 1805; *Hatcheria* Eigenmann, 1909; *Ituglanis* Costa and Bockmann, 1993; *Rhizosomichthys* Miles, 1942; *Scleronema* Eigenmann, 1917; *Sylvinichthys* Arratia, 1998; *Trichomycterus* Valenciennes, 1832 (De Pinna, 1998, Ochoa et al., 2017, sensu stricto Katz et al., 2018); and *Cambeva* (Katz et al., 2018). They are widely distributed in the neotropical region, from Costa Rica to Patagonia, and on both sides of the Andes [2,3]; while most genera comprise 1–3 species each, with restricted geographic distributions, *Trichomycterus* and *Cambeva* have over 180 known species between them [4].

The presence of odontodes outside the oral cavity organized in patches in the opercular region allows these animals to anchor to the substrate while in strong currents. These

dermal teeth may be responsible for the presence of these fish in lotic environments in the very high portions of river headwaters [2,3,5].

This group of catfishes has been widely investigated using ecological, morphological, and, more recently, molecular approaches, with integrative studies clarifying the relationships within the subfamily. Recently, a multilocus molecular analysis was employed to reconstruct the phylogenetic relationships of Trichomycteridae with an unprecedented level of sampling [6]. The authors recovered the nonmonophyletic status of *Trichomycterus*, showing that some groups were more related to *Eremophilus*, *Bullockia*, and *Scleronema*. However, their analysis did not include the type species of the genus, *T. nigricans*, nor did it consider the diversity of species of *Trichomycterus* from Atlantic coastal drainages.

Later, one study used 50 terminal taxa of *Trichomycterus* and 21 terminal taxa belonging to other Trichomycterinae genera (including *Bullockia*, *Eremophilus*, *Ituglanis*, and *Scleronema*) to perform the most comprehensive taxonomic revision of *Trichomycterus* to date based on phylogenetic analyses of molecular and morphological characters [7] and confirmed the structuring of two well-supported lineages and the northern lineage (which includes the type species *T. nigricans*) corresponding to *Trichomycterus* sensu stricto. Meanwhile, the authors assigned the southern lineage (species from Paraná, São Francisco, Ribeira de Iguape, and Uruguay river basins) to the new genus *Cambeva*. The species from the Guianas and the Andes were not investigated in the previous studies [6,7], but rather indicated as an opportunity for future research efforts.

The species of *Trichomycterus* from the Pakaraima Mountains in the Guiana Shield were investigated [8], and the authors found a high diversity of species (*T. conradi*; *T. guianense*; *T. cf. guianense*; and *Trichomycterus* sp., a spotted form different from the others) living in sympatry and syntopy in a restricted area.

Nevertheless, uncertainties and confusion remain concerning the morphological recognition of the species within *Trichomycterus* and the validity of the classification of certain species. The large number of species associated with high endemism or restricted distributions, sympatry among species, and incipient diagnoses [2,9] hamper efforts to resolve species classifications and their interrelationships.

Given that cladogenic events like geomorphological transformations may imprint genetic signatures in the lineages, it may be possible to determine whether the family's geographical structure is associated with the cladogenesis of lineages. Because *Trichomycterus* arose very recently (~18 mya) [6] and the species tend to occupy the headwaters of mountain watersheds, with frequent syntopy and geographically restricted species [2,3,5], speciation processes may be heavily affected by such events.

Given the group's recent cladogenesis [6], there is a high species diversity, with most species geographically restricted. Meanwhile, headland capture events may have greatly influenced their distribution. We therefore aimed to explore an unprecedented collection of specimens to investigate the divergence in the lineages of representatives of *Trichomycterus* in a wide geographic area. We expect our study to deepen efforts to understand the structural patterns of species distribution within Trichomycterinae in South America.

## 2. Materials and Methods

### 2.1. Mapping Patterns of Species Distribution

A survey of the type locality of *Trichomycterus nigricans* Valenciennes 1832 was performed to understand the general pattern of distribution across altitude regions. For the species survey, the FishBase–Catalogue of Life [1] was consulted. To obtain information on the type localities of species in Trichomycterinae, the original descriptions of the species and complementary studies were utilized to enhance the accuracy of the type locality, particularly in cases where the geographic data of the holotype was not provided or in very old descriptions. If a study did not contain the geographic coordinates, these data were obtained by referencing Google Earth® and following the description of the locality. For each species, the basin, city, state, country, latitude, longitude, holotype code, and species

authority were recorded (Table S1). Finally, a map that correlates to the species distribution to the region's altitude was created.

### 2.2. Sampling

Our analysis involved georeferenced 566 sequences (see Table S2 for Genbank acession number), being 33 species of Trichomycterinae, and 364 sequences of *Trichomycterus* sensu stricto (including sequences of the type species of the genus *T. nigricans* Valenciennes 1832). Of the 324 Brazilian sequenced catfishes (323 from *Trichomycterus* and one from *Cambeva*), 182 were collected by our team, and 142 were obtained from fish collections. We used four sequences as an external group, two from *Listrura* spp. (Glanapteryginae), and two from *Trichogenes longipinnis* Britski and Ortega 1983 (Trichogeninae). We also used sequences from Genbank® of cis- and trans-Andean species of Trichomycterinae, the locations of which ranged from the Isthmus of Panama in the north to Patagonia at the extreme southern limit of the subfamily's range (Table S2).

The samples from the Brazilian Atlantic coast comprised 13 sequences of *Cambeva* spp., four of *Scleronema* spp., and 14 of *Ituglanis* spp. The Andean samples consisted of 125 sequences of *Trichomycterus areolatus* Valenciennes 1846, 10 of *Hatcheria macraei* Girard 1855, and 12 of *Bullockia maldonadoi* Eigenmann 1928. The sequences from the northern Amazon basin consisted of 14 sequences of the *T. guianense* group Eigenmann 1909 from the Guiana Shield, eight from the Magdalena basin, and one *Trichomycterus* sp. from the Tapajós basin.

Fieldwork was carried out between June 2013 and October 2014 on expeditions that revisited the type localities of the eastern Brazilian species between the Jequitinhonha River basin in the north and the Itabapoana River basin in the south. Specimen sampling procedures followed a standard protocol [10], mainly using trawl nets. Captured specimens were anesthetized by submersion in eugenol solution [11] and the guidelines for fish euthanasia [12]. Tissue samples of the right flank muscle were obtained from specimens fixed in 90% ethanol and preserved in absolute ethanol for molecular analyses. For morphological comparisons, specimens were fixed in formalin for 15 days and then transferred to 70% ethanol. All specimens were registered in the fish collection at the Instituto Nacional da Mata Atlântica (INMA), formerly the Museu de Biologia Professor Mello Leitão, in Santa Teresa, Espírito Santo, Brazil.

### 2.3. Sequencing and Phylogenetic Analysis

DNA samples were extracted [13] and quantified using a NanoDrop ND-1000 spectrophotometer (Thermo Scientific™). For the in vitro amplification of the mitochondrial gene cytochrome b (*cytb*), we used the primers CytbSiluF and CytbSiluR [14]. Amplicons were purified by adding 1 μL Exosap to each 10 μL of PCR product. Sequencing reactions were performed using the Big Dye Terminator Cycle Sequencing Kit, following the manufacturer's protocol. The 332 samples were sequenced in an ABI310 automatic sequencer (Applied Biosystems) at the Núcleo de Biodiversidade Genética Luiz Paulo de Souza Pinto (Nubigen), at the Universidade Federal do Espírito Santo (UFES). The sequences were aligned using Geneious v. 7.1.3 [15]. The saturation test and concatenation of the sequences with identical haplotypes were performed in DAMBE5 [16,17]. The maximum likelihood (ML) tree was generated in MEGA version 7 [18], with 1000 bootstrap replications, "General Time Reversible" model, using gamma distribution and invariant sites, according to results suggested by jModelTest [19].

The Bayesian inference (BI) analysis was conducted in BEAST 2.5 [20], programmed with a relaxed lognormal model and uncorrelated with estimated rate with the ucld.mean parameter set (prior distribution lognormal, initial value of 0.009, mean of 0.009, and log-stdev of 0.0001) and the 0.5 interval uniform distribution, while the other parameters used the default settings. The length of the Markov chain Monte Carlo (MCMC) was 10 million generations, with sampling every 1000 generations. ESS (>200) values were checked using Tracer v. 1.7.1 [21], with the initial 2000 trees discarded as a burn-in period,

and the evolution model was found using jModelTest [19]. The main clades recovered by previous studies [6–8,22] were selected as monophyletic in the analyses: *T. guianense*, *T.* cf. *guianense*, and *T. conradi* [6,8]; genus *Cambeva* [7]; *Scleronema + Cambeva* [6], and Trichomycterinae [6,22]. Clade robustness was assessed through a bootstrap analysis and clade reliability was measured, considering strong support as >70% and moderate support as 50–70% [23]. Branch support was obtained through posterior probabilities, while clade confidence was defined as either strong (>0.95) or moderate (0.85–0.95). We edited the final trees in FigTree v1.4 [21] and generated the map in ArcMap10.1 [24]. The intra- and intergroup distance percentages were calculated in MEGA version 7 [18] using the Kimura-2-parameters model [25], with rates among sites set to "gamma distributed", gaps/missing data treatment set as "pairwise deletion", and other parameters set to default. The boxplot graphic was performed with R 4.2.1 [26]. The molecular phylogenetic analysis was carried out to estimate divergence times using BI in BEAST 2.5 [20] with the same parameter as provided above. The dating of the branches was calibrated following the values previously known [6].

## 3. Results

Through a survey of the type localities of nominal species of 178 taxa of Trichomycterinae (including 168 of *Trichomycterus* and 10 of phylogenetically related genera), we observed that these catfishes occur mainly in fast-flowing mountain watersheds that flow into the Pacific and Atlantic Oceans (Figure 1) and that the catfishes were rare (or unrecorded) in the lentic habitats of neotropical wetlands such as those of the Amazon and Pantanal regions (Table S1).

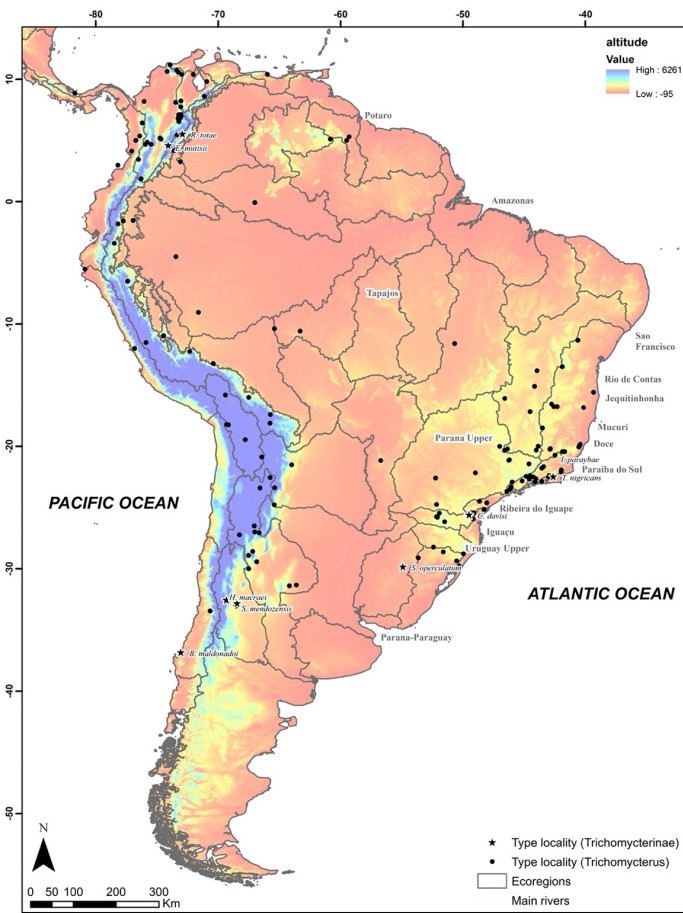

**Figure 1.** Type localities of 168 species of Trichomycterinae. Distribution of type localities of the 168 species nominally allocated in *Trichomycterus* in relation to altitude in the neotropical region. Stars indicate the type locality of each type of species of the genera to Trichomycterinae. Detailed localities are shown in Table S1, available only in the online version.

The present study presents well-supported evidence of genetic partitioning with geographic structuring among representatives of *Trichomycterus*, with an unprecedented level of sampling of specimens across geographical locations, using the data from the specimens sequenced in the present study (n = 324) and Trichomycterinae sequences from other molecular studies [6–8,27]. These sequences comprise thirty-three species, with twenty-four belonging to *Trichomycterus* sensu stricto, seven to *Cambeva*, two to *Scleronema*, and three to *Hatcheria*. We propose seven additional clades that are phylogenetically distinct from the known genera of Trichomycterinae.

### 3.1. Relationships among Major Clades

The dataset of 570 georeferenced sequences of Trichomycteridae with the 999-bp fragment of the *cytb* gene generated 409 haplotypes, with no saturation. We used the "General Time Reversible" method, with gamma distribution and invariant sites (GTR + I + G), as an evolution model.

We found divergence values from 17.8 to 20.5% between the three subfamilies (Trichomycterinae, Glanapteryginae, and Trichogeninae), with the lowest divergence between Trichomycterinae and Glanapteryginae, and the highest between Trichogeninae and Glanapteryginae.

The ML and BI trees were identical for Trichomycterinae, recovering ten strongly supported clades (clades A–J), structured geographically into two east-west major clades (EW major clade; Figure 2 and Figures S1–S3), with a divergence of 12.5% (intraeast = 7%; intrawest = 5%). The distances between clades A and J ranged from 9.7 to 15.7% (Table 1).

**Table 1.** Percentage of genetic distance pairwise between genera of Trichomycteridae. Bolded values on the diagonal are intraclade distances. Values below the diagonal (dark grey) represent the genetic distance between the respective clades. Values above the diagonal (blue) are the standard error of the distance indices.

| %   | A     | B     | C     | D     | E     | F     | G     | H     | I     | J     | K     | L     |
|-----|-------|-------|-------|-------|-------|-------|-------|-------|-------|-------|-------|-------|
| A   | **0.067** | 0.016 | 0.017 | 0.022 | 0.021 | 0.020 | 0.019 | 0.022 | 0.023 | 0.029 | 0.034 | 0.031 |
| B   | 0.103 | **0.054** | 0.018 | 0.017 | 0.021 | 0.020 | 0.020 | 0.020 | 0.022 | 0.028 | 0.033 | 0.030 |
| C   | 0.111 | 0.113 | **0.051** | 0.022 | 0.020 | 0.019 | 0.019 | 0.022 | 0.024 | 0.028 | 0.035 | 0.029 |
| D   | 0.129 | 0.105 | 0.120 | **0.072** | 0.026 | 0.024 | 0.022 | 0.025 | 0.025 | 0.029 | 0.033 | 0.035 |
| E   | 0.125 | 0.122 | 0.110 | 0.135 | **0.032** | 0.018 | 0.017 | 0.022 | 0.022 | 0.026 | 0.032 | 0.037 |
| F   | 0.128 | 0.123 | 0.105 | 0.131 | 0.101 | **0.050** | 0.016 | 0.019 | 0.021 | 0.025 | 0.029 | 0.031 |
| G   | 0.124 | 0.124 | 0.106 | 0.128 | 0.097 | 0.098 | **0.064** | 0.020 | 0.020 | 0.026 | 0.031 | 0.033 |
| H   | 0.140 | 0.128 | 0.137 | 0.151 | 0.128 | 0.126 | 0.124 | **0.095** | 0.023 | 0.027 | 0.035 | 0.035 |
| I   | 0.143 | 0.134 | 0.131 | 0.150 | 0.120 | 0.129 | 0.117 | 0.137 | **0.031** | 0.028 | 0.034 | 0.040 |
| J   | 0.156 | 0.153 | 0.152 | 0.154 | 0.131 | 0.136 | 0.147 | 0.157 | 0.165 | **n/c** | 0.039 | 0.040 |
| K   | 0.179 | 0.175 | 0.192 | 0.181 | 0.176 | 0.162 | 0.176 | 0.191 | 0.185 | 0.201 | **0.046** | 0.041 |
| L   | 0.177 | 0.176 | 0.168 | 0.190 | 0.192 | 0.171 | 0.184 | 0.191 | 0.221 | 0.205 | 0.205 | **0.000** |

Legend: A: *Trichomycterus* sensu stricto (Trichomycterinae). B: *Cambeva* spp. (Trichomycterinae). C: *Scleronema* spp. (Trichomycterinae). D: *T.* cf. *knerii* and *T. sandovali* (Trichomycterinae). E: *Hatcheria* spp. (Trichomycterinae). F: *T. striatus* and *T. straminius* (Trichomycterinae). G: *T. guianense, T.* cf. *guianense* and *T. conradi* (Trichomycterinae). H: *Ituglanis* spp. (Trichomycterinae). I: *T. transandianus* (Trichomycterinae). J: *Trichomycterus* sp. "tapajos" (Trichomycterinae). K: *Listrura* spp. (Glanapteryginae). L: *Trichogenes longipinnis* (Trichogeninae).

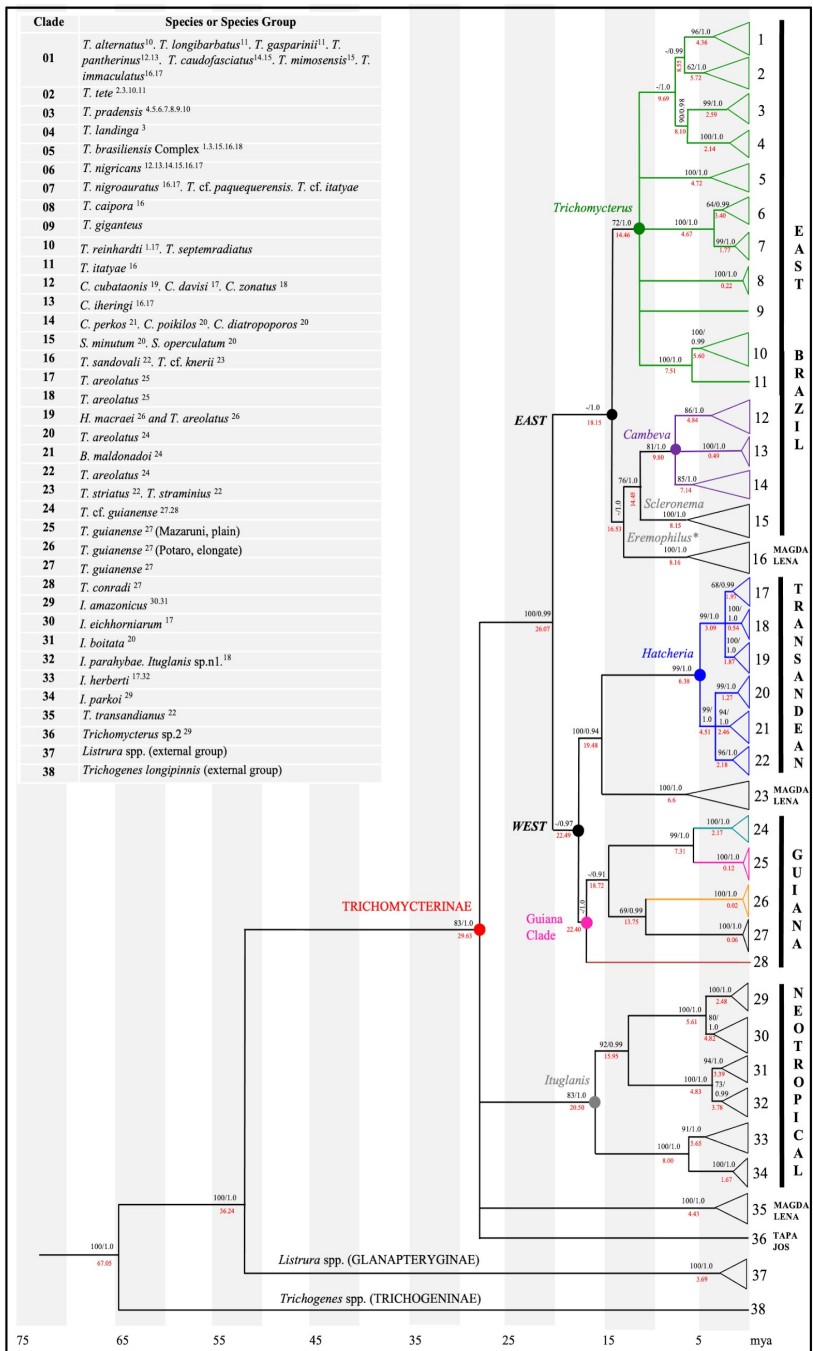

**Figure 2.** Phylogenetic tree of Trichomycterinae based on 566 cytochrome b sequences (405 haplotypes) with 999-bp. Topology of the tree refers to the analysis generated by Bayesian inference. Values above each branch correspond to the branch consistency index (ML bootstrap/BPP). The minus signal (-) corresponds to support lower than 60 in ML and 0.90 in BI. The red numbers (below the branch) represent the time of clade divergence (in millions of years ago). Inset box: superscript numbers refer to species by clade and their distribution by ecoregion: [1]São Francisco; [2]Rio de Contas; [3]Jequitinhonha; [4]Buranhém; [5]Jucuruçu; [6]Mucuri; [7]Itaúnas; [8]São Mateus; [9]Barra Seca; [10]Doce; [11]Piraquê-Açu; [12]Santa Maria da Vitória; [13]Jucu; [14]Itapemirim; [15]Itabapoana; [16]Paraíba do Sul; [17]Paraná-Paraguay; [18]Ribeira do Iguape; [19]Itapocu; [20]Jacui; [21]Uruguay; [22]Magdalena (MAG); [23]Orinoco; [24]Central region of trans-Andean Chile basins; [25]South region of trans-Andean Chile basins; [26]Southern end of trans-Andean Chile basins; [27]Potaro. Guiana; [28]Mazaruni, Guiana; [29]Tapajós (Amazonas; TAP); [30]Madeira (Amazonas); [31]Jari (Amazonas); and [32]Araguaia.

For the east major clade, *Trichomycterus* sensu stricto (clade A), *Cambeva* (clade B), *Scleronema* (clade C), and *T. sandovali* + *T. knerii* (clade D) were recovered, with genetic distances of 10.3 to 12.9%, and intrageneric distances of 5.1 to 7.2%. For the west major clade, *Hatcheria macraei* + *Bullockia maldonadoi* + *T. alternatus* (clade E: genus *Hatcheria*), *T. striatus* + *T. straminius* (clade F), and *T. guianense* + *T. conradi* (clade G) were found, with genetic distances of 9.7 to 10.1%, and intrageneric distances of 3.2 to 6.4% (Table 1). *Ituglanis* spp. (clade H) was recovered as a monophyletic genus, in addition to two other clades less related to the other Trichomycterinae: *T. transandianus* (clade I) and *Trichomycterus* sp. (clade J: from Tapajós basin, sensu [6]). Based on the genetic divergence, geographic structure, and distribution (Figure 3), we analyzed these 10 clades (A–J) as potential genera or potential new genera arrangements.

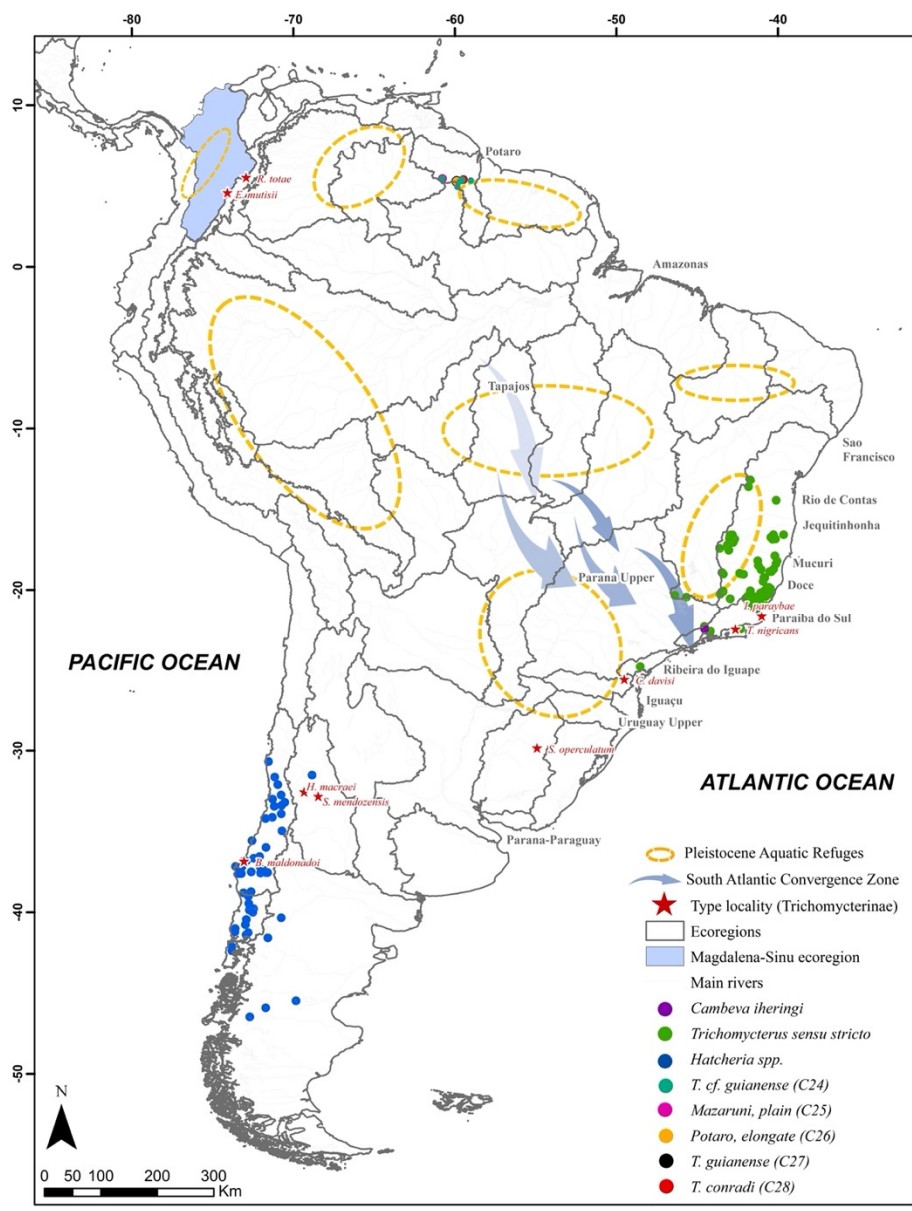

**Figure 3.** Distribution of the georeferenced lineages of the Trichomycterinae subfamily in the neotropical region. Geographic area divided in ecoregions, sensu [28]. Stars indicate the type locality of each type species within each subfamily. Orange dotted circles representing aquatic Pleistocene refugia [29]. Note: many areas do not have the coordinate of the sample. Instead, authors cited the basin of occurrence (for details see Table S1). This is the case of highlights on the map of Magdalena-Sinu ecoregion.

*3.2. East Major Clade*

Included nominal species: *Trichomycterus alternatus, T. longibarbatus, T. pantherinus, T. caudofasciatus, T. immaculatus, T. mimosensis, T. tete, T. pradensis, T. landinga, T. brasiliensis, T. brunoi, T. candidus, T. macrotrichopterus, T. pirabitira, T. rubiginosus, T. nigricans, T. nigroauratus, T.* cf. *paquequerensis, T.* cf. *itatyae, T. caipora, T. giganteus, T. reinhardti, T. septemradiatus, T. itatyae, Cambeva cubataonis, C. davisi, C. zonatus, C. iheringi, C. perkos, C. poikilos, C. diatropoporos, Scleronema minutum, S. operculatum, T. sandovali,* and *T.* cf. *knerii.*

The clade comprising *Trichomycterus, Scleronema, Cambeva,* and *Eremophilus* was recovered as monophyletic, with *Trichomycterus* as sister to *Cambeva + Scleronema* (TCS), and *Eremophilus* as sister to the TCS clade.

The clade *Trichomycterus sandovali* and *T. knerii* (subclade 16) from the Magdalena Ecoregion was recovered as sister to the *Cambeva + Scleronema* clade, the unique lineage from the north included in the lineages from eastern Brazil, comprising *Trichomycterus* sensu stricto, *Cambeva, Scleronema,* and subclade 16.

Beyond the distinct genus status, the Trichomycterinae from eastern Brazil were grouped into four endemic and phylogenetically related genera and subdivided into 15 subclades (1–15; Figure 2), as follows: *Trichomycterus* sensu stricto (subclades 1–11), *Cambeva* (subclade 12–14), and *Scleronema* (subclade 15). *Cambeva* and *Scleronema* were recovered as sister genera (ac. 16.05 mya), with a genetic distance of 12.7%. The clade *Trichomycterus* sensu stricto showed a genetic distance of 13.1% to the clade *Cambeva + Scleronema.* Among the four genera, *Trichomycterus* has the northernmost distribution within the subfamily and coalesced from the others at approximately 18.02 mya.

The distribution of *Scleronema* lies to the south of *Cambeva. Trichomycterus* and *Cambeva* are sympatric in the headwaters of the Paraíba do Sul, São Francisco, and Paraná basins, where *Trichomycterus* subclades 10–11 are endemics.

The main eastern Brazilian clades (*Trichomycterus* sensu stricto) + (*Cambeva + Scleronema*), sister of clade D (subclade 16, comprising *T. sandovali* from Magdalena and *T.* cf. *knerii* from Orinoco), show an estimated coalescence time of 27.57 mya. Subclade 16 has an intrageneric distance of 8.8%, differing from the eastern Brazilian clade by 13.4%.

The relationship of the other genera was not recovered because they were recovered as polyphyletic. The clade *Ituglanis* was recovered with high support and widely distributed in the neotropical region.

### 3.2.1. Trichomycterus Sensu Stricto—Clade A

Included species: *T. alternatus, T. longibarbatus, T. pantherinus, T. caudofasciatus, T. immaculatus, T. mimosensis, T. tete, T. pradensis, T. landinga, T. brasiliensis, T. brunoi, T. candidus, T. macrotrichopterus, T. pirabitira, T. rubiginosus, T. nigricans, T. nigroauratus, T.* cf. *paquequerensis, T.* cf. *itatyae, T. caipora, T. giganteus, T. reinhardti, T. septemradiatus,* and *T. itatyae.*

We recovered *Trichomycterus* sensu stricto within two groupings from eastern Brazil with high genetic divergence of 9.8%. Across all 11 subclades in the clade *Trichomycterus* sensu stricto (subclades 1–11, Figure 2), pairwise genetic distances ranged between 4.3% and 11.3% (Table S3). One grouping (subclades 1–9) includes the type species *T. nigricans*; the second (subclades 10–11) is located at the southern limit of *Trichomycterus* sensu stricto.

The representatives of the clade occur throughout the coastal drainages of the Brazilian Atlantic coast and the Upper Paraná (ACUP) watershed of the southern Brazilian Shield. Its range is bordered by the Rio de Contas basin to the north and Ribeira do Iguape to the south, extending through ecoregions fully or partially within the Atlantic Forest. The western limits of the distribution correspond to the divide with the Rio São Francisco basin. Species of Trichomycterinae are generally limited to the headwaters and tributaries of river basins.

Subclade 1 has an intraclade distance of 2.9% and comprises seven taxa from the southern distribution of *Trichomycterus.* The subclade is only present in the Rio Doce basin; it comprises *T. alternatus* (Doce), *T. longibarbatus* (Piraquê-Açu), *T. pantherinus* (Santa Maria da Vitória and Jucu), *T. caudofasciatus* (Itapemirim and Itabapoana), *T. mimosensis*

(Itabapoana), *Trichomycterus* sp. (Paraíba do Sul), and *T. zonatus* (Ribeira do Iguape). Subclade 2 is represented by *T. tete* from the Rio de Contas basin and *T. jequitinhonhae* from the Jequitinhonha basin in the Espinhaço mountains. Subclade 2 diverged from subclade 1 with a genetic distance of 6.9%. Subclade 3 consists of only one species, *T. pradensis*, which is widely distributed from the northern Doce basin to the southern Jucuruçu watershed. Subclade 4 diverged from subclade 3 with a genetic distance of 5.6%. Subclade 4 comprises *T. landinga*, endemic to the Jequitinhonha River basin. Subclades 1 to 4 form a clade and cover the entire distribution of *Trichomycterus* sensu stricto. Subclade 5 comprises the *Trichomycterus brasiliensis* species complex with an intraclade distance of 3%. The subclade's distribution extends to the São Francisco, Jequitinhonha, Itabapoana, and Paraíba do Sul basins, reaching the Atlantic Forest biome in its western limits. Subclade 6 diverged from subclade 7 with a genetic distance of 4.3% and occurs in sympatry in the Paraíba do Sul and Paraná–Paraguay basins. Subclade 6 consists exclusively of the type species of the genus, *T. nigricans*, which occurs in the Santa Maria da Vitória, Jucu, Itapemirim, Itabapoana, and Paraíba do Sul basins, overlapping with the distribution of subclade 1 at its southernmost extent. Subclade 7 consists of *T. nigroauratus*, *T.* cf. *paquequerensis*, and *T.* cf. *itatyae*, with a distribution in the Paraíba do Sul and Upper Paraná basins. Subclade 8 consists of *T.* cf. *caipora* from the Paraíba do Sul basin. Subclade 9 is represented by *T. giganteus*, occurring in Guanabara Bay, Rio de Janeiro.

Subclades 10 and 11 coalesced from subclades 1–9 of *Trichomycterus* sensu stricto at an estimated 14 mya, with the highest genetic distance from the other subclades in the genus.

Subclades 10 and 11 comprise endemic species from the upper Paraná (*T. septemradiatus*), upper São Francisco (*T. reinhardti*), and upper Paraíba do Sul (*T. itatyae*) watersheds, each of which is geographically isolated. The main overlapping area of sympatry among the subclades is the Paraíba do Sul basin (subclades 1, 6, 7, 8, and 11), a sympatric region between *Trichomycterus* sensu stricto and *Cambeva*, as it represents the southern limit of *Trichomycterus* and the northern limit of *Cambeva*.

### 3.2.2. Cambeva—Clade B

Included species: *Cambeva cubataonis*, *C. davisi*, *C. zonatus*, *C. iheringi*, *C. perkos*, *C. poikilos*, and *C. diatropoporos*.

*Cambeva* is represented by three subclades (subclades 12–14) with genetic distances between 7.5% and 9.7% and an intrageneric distance of 6.5% (Table 1). Subclade 12 comprises *C. cubataonis*, *C. davisi*, and *C. zonatus*, with a distribution that stretches across the Itapocu, Ribeira do Iguape, and Paranapanema (Paraná) basins, with a genetic distance of 3.9% within the subclade (Table S3). Subclade 13 is represented by *C. iheringi* with a range in the Paraíba do Sul and Paraná basins, in sympatry with *Trichomycterus* (subclades 1, 5, 6, 7, 8, 10, and 11). Subclade 14 is formed of *C. perkos*, *C. poikilos*, and *C. diatropoporos*, with a distribution including the Jacui and Upper Uruguay basins, representing the southern limit of the genus, a region in which it occurs in sympatry with *Scleronema*.

### 3.2.3. Scleronema—Clade C

Included species: *Scleronema minutum* and *S. operculatum*.

*Scleronema* (clade C, subclade 15) is represented by *S. minutum* and *S. operculatum* with an intraclade genetic distance of 6.1% (Table 1). Its distribution is limited to the Lower Uruguay basin, the southern limit of the eastern Brazilian clade. Subclade 15 was recovered as sister to the clade *Cambeva*, with a genetic distance of 12.7%. *Scleronema* occurs in sympatry with *Cambeva* in the Uruguay basin, at the southern edge of the latter's range.

### 3.2.4. Eremophilus—Clade D

Included species: *Trichomycterus sandovali* and *T.* cf. *knerii*.

Clade D (subclade 16) represented by *T. sandovali* and *T.* cf. *knerii* is known as the clade *Eremophilus* due to the previous recovered phylogenetic relationship [6], which includes *E. mutisii* as a member of the clade with the species mentioned above. The clade *Eremophilus*

is distributed in the Orinoco and Magdalena basins and is sister to *Cambeva + Scleronema*, both endemic genera to eastern Brazil.

### 3.3. West Major Clade

Included species: *Trichomycterus areolatus*, *Hatcheria macraei*, *Bullockia maldonadoi*, *T. striatus*, *T. straminius*, *T.* cf. *guianense*, *T. guianense*, and *T. conradi*.

The west major clade is represented by the genus *Hatcheria* (clade E, subclades 17–22), comprising *H. macraei*, *B. maldonadoi*, and *T. areolatus* from the central and southern regions of trans-Andean Chile; *T. striatus* and *T. straminius* (clade F, subclade 23), both from the Magdalena basin, and the Guiana clade formed by *T. guianense*, *T.* cf. *guianense*, and *T. conradi* (clade G, subclades 24–28) from the Potaro basin of the Essequibo ecoregion, corresponding to the Guiana Shield portion.

Hatcheria—Clade E

Included species: *Trichomycterus areolatus*, *Hatcheria macraei*, and *Bullockia maldonadoi*.

The trans-Andean Austral clade was recovered as a monophyletic clade with a reference intraclade distance of 3.8%, coalescing in two subclades (17–19 and 20–22) at an estimated 18.59 mya, with an intraclade genetic distance of 3.2% and an intersubclade genetic divergence of 6%. Representatives of *Trichomycterus areolatus*, *Hatcheria macraei*, and *Bullockia maldonadoi* are grouped in subclade 17–19 from the southernmost area of Chile (*Trichomycterus areolatus* and *Hatcheria macraei*), with an intrasubclade divergence of 2.5%. Subclade 20–22 is represented by catfishes from the central region of trans-Andean Chile and comprises *T. areolatus* and *Bullockia maldonadoi*, with an intrasubclade divergence of 3.2%. Specimens identified as *T. areolatus* from clades 17 and 18 diverged by 6.1% from the *T. areolatus* individuals from clades 20 and 22.

With respect to intrageneric divergence, it is reasonable to assume that *T. areolatus* and *B. maldonadoi* (subclade 21) should be allocated to *Hatcheria*. The divergence levels in subclades 17–22 are compatibles with distinct species of the same genus. Furthermore, the divergence and cladogenesis findings do not corroborate the validity of three genera. In our analysis, *Bullockia* is recognized as a junior synonym to *Hatcheria*. Because we found that *Trichomycterus* sensu stricto is restricted to species inhabiting eastern Brazil (subclades 1–11), *Trichomycterus areolatus* Valenciennes 1846 is therefore recognized as *Hatcheria areolata* n. comb. In our analysis, *Hatcheria* is recognized as a single clade and the only genus of trichomycterids inhabiting austral freshwater environments. From the delimitations of the lineages obtained through data on genetic divergence and phylogenetic cladogenesis, we identify six species of *Hatcheria*. From these same criteria, we recover *H. aerolata* as polyphyletic (subclades 17, 18, 20, and 22), but it was not possible to determine the clade to which the species should belong. Thus, *H. areolata* can belong to any of these four subclades, which can also include previously described southern species or even new species. However, as a previous study aimed to understand the biogeography of *T. areolatus* [30], and the most representative subclade is 17, it is reasonable to assume that the species *H. areolata* belongs to this clade.

### 3.4. Striatus—Clade F

Included species: *Trichomycterus striatus* and *T. straminius*.

Clade F (subclade 23) comprises *T. striatus* and *T. straminius*, with an intrageneric distance of 5%, endemic to the Magdalena basin. Clade *striatus* is sister to the clade *Hatcheria*, comprising cis-trans Andean species (*H. macraei*, *B. maldonadoi*, and *T. areolatus*), with a genetic distance of 9.8% in relation to this clade.

### 3.5. Guiana—Clade G

Included species: *Trichomycterus* cf. *guianense*, *T. guianense*, and *T. conradi*.

The clade Guiana is formed of *T. guianense*, *T.* cf. *guianense*, and *T. conradi* (clade G, subclades 24 to 28), with an intraclade genetic distance of 6.4%. It is distributed in



the Potaro basin of the Essequibo ecoregion, corresponding to the Guiana Shield portion. Interclade genetic distances with other genera vary from 9.7% (clade E: *Hatcheria* from trans-Andean portion) to 15.7% (clade J from Tapajós). These distances are compatible with genus-level differences, suggesting that this clade is a distinct genus.

### 3.6. Ituglanis—Clade H

Included species: *Ituglanis parkoi, I. herberti, I. boitata, I. parahybae, Ituglanis* sp. n1, *I. amazonicus*, and *I. eichhorniarum*.

Based on our analysis, *Ituglanis* is represented by six subclades (29–34) distributed throughout the neotropical region: *I. amazonicus* from Madeira and Jari in the Amazon basin (subclade 29), sister to *I. eichhorniarum* from Taquari and Aquidauana in the Paraguay and Paraná basins (subclade 30); *I. boitata* from Jacui in the Laguna dos Patos basin (subclade 31), sister to *I. parahybae* in the Macabu and Ribeira do Iguape basins (subclade 32); and *I. herberti* from Araguaia in the Tocantins basin and Piraí in the Paraná basin (subclade 33), sister to *I. parkoi* from Tapajós in the Amazonas basin (subclade 34; Table S2). The clade *Ituglanis* has an intrageneric distance of 9.5%. Interspecific distances vary between 4.3% and 13.1%, while intraspecific distances range from 1.6% to 4.6%. These high divergence rates among congeneric species indicate a need for revision since *Ituglanis* may represent more than one genus. Although the genus was recovered as monophyletic, it is distributed throughout the entire neotropical region, explaining the high divergence between more geographically distant species (Table S3).

### 3.7. Transandianus—Clade I

Included species: *Trichomycterus transandianus*.

Based on the genetic distance and geographic distribution of the monophyletic clades, two additional clades that have not been included in any known genus already delimited in the phylogeny were recovered: one from the Amazon and the other from the Magdalena basin (Figure S2). The two distinct lineages of Trichomycterinae herein recovered are termed clades I and J. Clade I (subclade 35), comprising *T. transandianus* from trans-Andean Magdalena, has a genetic distance to other genera of Trichomycterinae that ranges from 12% (clade E) to 16.5% (clade J) and an intraclade distance of 3.1%.

### 3.8. Tapajós—Clade J

Included species: *Trichomycterus* sp.2.

The second distinctive clade (J, subclade 36) comprises *Trichomycterus* sp.2 as defined previously [6] and consists of a single unnamed specimen, a catfish from the Tapajós basin (Amazonas). It has the greatest genetic distance from all the other clades, ranging from 13.1% (clade E) to 16.5% (clade I).

### 3.9. Referential Indices of Molecular Divergence for Trichomycterinae

Intrageneric divergences for Trichomycterinae, Glanapteryginae, and Trichogeninae ranged from 3.1% to 8.1% (except *Ituglanis*, where it was 10%). The genetic distance between the subfamilies varied from 17.8% to 20.5%, while it was between 9.7% and 15.7% across genera of Trichomycterinae (Tables 1 and 2; Figure 4). The divergence between species or species groups within genera ranged from 2.5% to 11.3%, except for those of *Ituglanis*, with higher values ranging from 4.3 to 13.1% (Table 2, Figure 4, Table S3).

**Table 2.** Percentage (%) of maximum, minimum, and mean genetic divergence between species, genera, and subfamilies of Trichomycterinae.

| Group | Minimum | Mean | Maximum | Mean (Intra) |
|---|---|---|---|---|
| *Trichomycterus s.s.* | 4.3 | 8.2 | 11.3 | |
| *Cambeva* | 6.3 | 8.6 | 10.9 | |
| *Hatcheria* | 2.5 | 6.3 | 11.5 | 8.5 |
| Guiana genus | 4.8 | 9.2 | 11 | |
| *Ituglanis* | 4.3 | 10.4 | 13.1 | |
| Genera into Trichomycterinae | 9.7 | 12.9 | 15.7 | 5.61 |
| Subfamilies (3) | 17.8 | 18.9 | 20.5 | 7.2 |

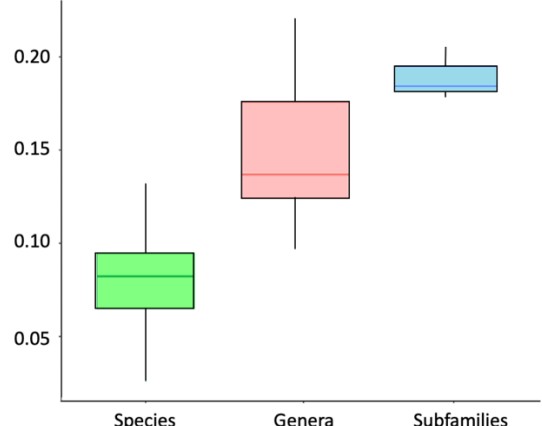

**Figure 4.** Boxplot of the genetic distance percentage interval in species (green), genera (red), and subfamilies (blue).

The lowest pairwise distance across genera of Trichomycterinae was 9.7%, between *Hatcheria* spp., from the trans-Andean region (clade E) and *T. guianense* + *T. conradi* from the Guiana Shield (clade G); while the highest distance (15.7%) was obtained between *Ituglanis* spp. (clade H) and *Trichomycterus* sp. from Tapajós (clade J).

Intergeneric indices within Trichomycterinae ranged from 9.7% to 16.5%, with the lowest value between *Hatcheria* (clade E) and species from Guiana (clade G) and the highest values between *T. transandianus* from Magdalena (clade I) and *Trichomycterus* sp. from Tapajós (clade J) (Table 1). We also noted that the most divergent lineages within Trichomycterinae were *T. transandianus* (Magdalena, trans-Andean), *Trichomycterus* sp. (Tapajós), and *Ituglanis*, with similar values (between 11.7% and 16.5%).

The distances between the main clades (A–J) suggest that they may represent different genera. However, more research is required to support such decisions, even though we have recovered clades with high support, high genetic distances (compatible with distances from previously defined and named genera), and geographic structuring. Our decisions were based on phylogenetic lineage support as well as the intraclade (A–J) divergences between 3.2% and 9.5% and interclade divergence from 9.7 to 15.7% of recognized genera.

The interclade genetic distances that grouped species into the same genus of Trichomycterinae varied between 2.5% and 11.5%, except in the case of *Ituglanis* species, which had values between 4.3% and 13.1% (Table S3). The pairwise genetic divergence of each genus with more than one species varied as follows: 10.3–15.6% for *Trichomycterus* sensu stricto; 10.5–15.3% for *Cambeva*; and 11.7–15.1% for *Ituglanis*. Since the genetic distances found in clade E (9.7–13.1%), comprising *Hatcheria macraei*, *Bullockia maldonadoi*, and *Trichomycterus areolatus*, and clade G (9.7–14.7%), comprising the species from Guiana,

*Trichomycterus guianense* and *T. conradi*, are compatible with the genus level, these clades may represent different genera.

The divergences among the subclades 17–22 of *Hatcheria* (as herein proposed) ranged from 2.5 to 5.7% (Table S3). The intrageneric values between clades in *Trichomycterus* sensu stricto ranged from 4.3% to 11.3%, with the highest genetic divergence values found between subclades 10 and 11 and all other subclades (mean pairwise divergence: 9.0% and 9.4%, respectively).

In each genus, interspecific divergence ranges were as follows: *Trichomycterus* sensu stricto from 4.3 to 11.3% (mean: 8.2%); *Cambeva* from 6.3 to 10.9% (mean: 8.6%); *Ituglanis* from 4.3 to 13.1% (mean: 10.4%); *Hatcheria* from 2.5 to 11.5% (mean: 6.3%), and clade Guiana from 4.8 to 11% (mean: 9.2%). Notably, the genetic distances in clades E and G are compatible with the genus level, suggesting that they may represent new genera. Specifically, clade E, comprising *Hatcheria macraei*, *Bullockia maldonadoi*, and *Trichomycterus areolatus*, had interspecific genetic divergence between 2.5% and 11.5% (mean: 6.3%), while clade G, comprising species from Guiana, as *Trichomycterus guianense* and *T. conradi*, had genetic distances ranging from 4.8 to 11% (mean: 9.2%).

We therefore propose that distances as high as 17.8–20.5% should be used to delimit different subfamilies, while values of 9.7–15.7% would separate genera within the same subfamily (Tables 1 and 2), and interspecific variation within a genus would have values of 2.5–11% (Table S3).

It is important to mention that genetic divergence per se is not sufficient to delimit taxa. However, since we found that the genetic divergence was also associated with geographically structured clades, our findings indicate a need to revisit species classification and occurrence. Furthermore, they suggest a considerable impact of landscape factors on the genetic signatures of these lineages. Our reference values of intra- and interspecific variation do not encompass the ranges found within *Ituglanis* since its range represents an outlier in relation to the other genera of Trichomycterinae.

Our study established reference values of genetic diversity for one mitochondrial gene at the subfamily, genus, and species levels representing the first evaluation of its kind for *Trichomycterus*. Subfamilies in the Trichomycteridae diverged by 21–24%, genera in Trichomycterinae by 11–20%, and congeneric species by 3–11%. Intraspecific distances ranged between 0.4% and 8.8%.

## 4. Discussion

All eastern Brazilian genera have been recovered as monophyletic, as follows: *Trichomycterus* sensu stricto as the sister clade of *Cambeva* + *Scleronema* and *T. sandovali* + *T.* cf. *knerii* (subclade 16 or *Eremophilus mutisii*) as the sister clade to the former group, *T.* cf. *knerii*, which occurs in the Magdalena and Orinoco basins (cis-Andean portion). This finding provides valuable evidence for the phylogenetic relationships between species from eastern Brazil and those from Magdalena and Orinoco, a highly diverse region for Trichomycterinae. Also, the monophyly of Trichomycterinae, *Ituglanis*, and *Hatcheria* was recovered, and due to the number and location of samples, the geographic distribution of the subfamily's genetic diversity throughout South America was characterized, with particular emphasis on the distribution limits of lineages from eastern Brazil.

Our potential confirmation of the delimitation of *Trichomycterus* sensu stricto to species from eastern Brazil underscores the urgency and necessity to revise the classification of species that are still nominally considered as being representatives of *Trichomycterus*, recorded in other geographical ranges (Table 3).

**Table 3.** Revisiting the classification of the species included in the genus. See Table S2 for detailed information on the number of GenBank specimens used in each reference. * *Eremophilus mutisii* multilocus sequences were recovered from the clade with *T. sandovali* and *T.* cf. *knerii* in previous papers [6,7].

| Genus/Clade | Subclade | Species | Distribution |
|---|---|---|---|
| *Trichomycterus* (A) | 1 | *Trichomycterus alternatus, T. longibarbatus, T. pantherinus, T. caudofasciatus, T. immaculatus, T. mimosensis* | Contas, Jequitinhonha, Barra Seca, São Mateus, Doce, Timbui, Piraque-Açu, Santa Maria da Vitoria, Jucu, itapemirim, itabapoana, Paraiba do Sul, Ribeira do Iguape |
| | 2 | *Trichomycterus tete* | Contas, Jequitinhonha, Piraque-Açu |
| | 3 | *Trichomycterus pradensis* | Buranhem, Jucuruçu, Mucuri, itaunas, Barra Seca, Doce, Jucu |
| | 4 | *Trichomycterus landinga* | Jequitinhonha |
| | 5 | *Trichomycterus brasiliensis, T. brunoi, T. candidus, T. macrotrichopterus, T. pirabitira, T. rubiginosus* | Itabapoana. Jequitinhonha. São Francisco. Parana |
| | 6 | *Trichomycterus nigricans* | Santa Maria da Vitoria. Jucu, itapemirim, itabapoana. Paraiba do Sul |
| | 7 | *Trichomycterus nigroauratus, T.* cf. *paquequerensis, T.* cf. *itatyae* | Paraiba do Sul. Parana |
| | 8 | *Trichomycterus caipora* | Paraiba do Sul |
| | 9 | *Trichomycterus giganteus* | Baia de Guanabara |
| | 10 | *Trichomycterus reinhardti, T. septemradiatus* | Parana. São Francisco |
| | 11 | *Trichomycterus itatyae* | Paraiba do Sul |
| *Cambeva* (B) | 12 | *Cambeva cubataonis, C. davisi, C. zonatus* | Itapocu, Ribeira do Iguape, Paranapanema |
| | 13 | *Cambeva iheringi* | Paraiba do Sul, Parana |
| | 14 | *Cambeva perkos, C. poikilos, C. diatropoporos* | Jacui, Uruguay |
| *Scleronema* (C) | 15 | *Scleronema minutum, S. operculatum* | Uruguay |
| D | 16 | *Trichomycterus sandovali, T.* cf. *knerii* | Magdalena, Orinoco |
| *Hatcheria* (E) | 17 | *Trichomycterus areolatus* | Transandinean (South) |
| | 18 | *Trichomycterus areolatus* | Transandinean (South) |
| | 19 | *Hatcheria macraei, Trichomycterus areolatus* | Transandinean (South) |
| | 20 | *Trichomycterus areolatus* | Transandinean (South) |
| | 21 | *Bullockia maldonadoi* | Transandinean (South) |
| | 22 | *Trichomycterus areolatus* | Transandinean (South) |
| F | 23 | *Trichomycterus striatus, T. straminius* | Magdalena |
| Guiana Clade (G) | 24 | *Trichomycterus* cf. *guianense* | Potaro, Guyana |
| | 25 | *Trichomycterus guianense* (Mazaruni, plain) | Potaro, Guyana |
| | 26 | *Trichomycterus guianense* (Potaro, elongate) | Potaro, Guyana |
| | 27 | *Trichomycterus guianense* | Potaro, Guyana |
| | 28 | *Trichomycterus conradi* | Potaro, Guyana |

**Table 3.** *Cont.*

| Genus/Clade | Subclade | Species | Distribution |
|---|---|---|---|
| *Ituglanis* (H) | 29 | *Ituglanis parkoi* | Tapajós–Amazonas basin |
| | 30 | *Ituglanis herberti* | Araguaia–Tocantins and Pirai–Paraná basins |
| | 31 | *Ituglanis boitata* | Jacui-Laguna dos Patos basin |
| | 32 | *Ituglanis parahybae, ituglanis* sp.n1 | Macabu and Ribeira do Iguape basins |
| | 33 | *Ituglanis amazonicus* | Mamoré–Madeira basin |
| | 34 | *Ituglanis eichhorniarum* | Taquari and Aquidauana –Paraguay, Paraguay–Paraná basins |
| I | 35 | *Trichomycterus transandianus* | Magdalena |
| J | 36 | *Trichomycterus* sp.2 (Tapajos) | Tapajos, Amazonas |
| *Listrura* (K) | 37 | *Listrura nematopteryx, L. picinguabae* | Neotropical |
| *Trichogenes* (L) | 38 | *Trichogenes longipinnis* | Rio de Janeiro and São Paulo States |

*4.1. The Major Clades in the Trichomycterinae*

Given that *Trichomycterus* sensu stricto, *Cambeva*, and *Scleronema* have been recognized as phylogenetically close but distinct genera, supported by ample morphological and genetic evidence [7], we anticipated that the divergences between other genera within the subfamily would be comparable to the values of 12.4% to 15.5% calculated for the clade. We also calculated the between-species distances within the *Trichomycterus* sensu stricto clades, ranging from 5.0% to 11.2%. In contrast, the genetic differences we observed between the trans-Andean Austral taxa were between 3.1% and 6.8%, which is incompatible to the notion of distinct genera.

4.1.1. Clade *Trichomycterus* Sensu Stricto

*Trichomycterus* sensu stricto is recognized as an assemblage with species from the northeastern Atlantic Forest ecoregion and adjacent basins. The intraclade relationships between species of *Trichomycterus* are out of the scope of the present study. *Trichomycterus* sensu stricto is a highly supported monophyletic group, geographically structured in eastern Brazil with a range bounded in the north by the Rio de Contas basin in Bahia state, the São Francisco basin in the west, and the Ribeira do Iguape river basin in the south. This range represents a distribution from 10° S to 25° S and from 40° W to 50° W. *Trichomycterus nigricans* is the type of species, and the type of locality is in a drainage of Rio de Janeiro state, Brazil [31]. The genus *Trichomycterus* is attributed solely to species occurring in eastern Brazil.

Until recently, species from southeastern and southern Brazil were considered part of *Trichomycterus*. Nevertheless, *Cambeva* has now been identified as a distinct genus with a geographic distribution that overlaps with *Trichomycterus* sensu stricto in the upper São Francisco, Paraná, and Paraíba do Sul basins [6]. The three clades from the Brazilian Shield [6] were also recovered: one corresponding to *Trichomycterus* with a northern distribution covering the Paraná, Paraíba do Sul, São Francisco, Itabapoana, Jucuruçu, and Doce basins; another to *Cambeva* in the southeastern portion of the distribution of *Trichomycterus*; and the third as *Scleronema* in lower Uruguay. In our analysis, we found that *Trichomycterus* sensu stricto and *Cambeva* occur in sympatry in the Paraíba do Sul, Upper Paraná, and upper São Francisco basins, highlighting the significance of this region in the biogeographic history and cladogenesis of the lineages within the eastern Brazilian Shield. The discovery of a potential new endemic sister genus of *Trichomycterus* sensu stricto underscores the remarkable diversity of Trichomycterinae in this area. Additionally, the coexistence of the established sister genera *Cambeva* and *Scleronema* in the lower Paraná–Uruguay further emphasizes the region's significance in hosting diverse Trichomycterinae species.

The genus *Trichomycterus* is still cited in the literature as having 166 species [29]. These numbers, however, do not consider the reallocation of species to *Cambeva*. Further efforts are required to diagnose and classify approaches since future studies aiming to understand the phylogeny of *Trichomycterus* or *Cambeva* must deal with samples whose identification has not been updated in the databases. This is particularly important given that studies are often based on samples with few specimens and low representativeness in terms of species and geographic areas.

In relation to *Trichomycterus alternatus* species group (Clade 1), we found that *T. alternatus* is part of a species complex we refer to as subclade 1 of *Trichomycterus* sensu stricto, comprising *T. longibarbatus*, *T. gasparinii*, *T. pantherinus*, *T. caudofasciatus*, and *T. mimosensis*. However, the molecular differences between the species within the complex are unclear. For example, *T. gasparinii* is indistinguishable from *T. longibarbatus*, and *T. mimosensis* remains indistinguishable from *T. caudofasciatus*. The low intraclade genetic distance within the species complex warrants future analysis to assess its validity.

The low genetic diversity associated with many valid species highlights the problem of the genus. Therefore, using a 2% genetic distance threshold may overestimate the number of species in Trichomycterinae, which cannot always be distinguished morphologically. By applying a conservative lumper view to delimit species genetically [32], we propose a threshold level of interspecific molecular divergence of at least 3%, as this value is sufficient to reveal different monophyletic species without creating morphological problems in distinguishing them. If a 2% threshold is used to validate molecular divergence in *T. alternatus*, *T. longibarbatus*, *T. gasparinii*, *T. pantherinus*, *T. caudofasciatus*, *T. immaculatus*, and *T. mimosensis*, we could likewise propose at least 13 new morphologically indistinguishable species.

In a previous study [22] the clade *immaculatus* was recovered as sister to the clade *brasiliensis*, which correspond to subclades 1 and 5 in the present study, respectively. We expanded the understanding of the relationship of the clade *immaculatus* sensu [22], here referred to as subclade 1 (*T. alternatus*, *T. longibarbatus*, *T. gasparinii*, *T. pantherinus*, *T. caudofasciatus*, *T. immaculatus*, and *T. mimosensis*), with other clades, positioning it as part of the same clade as subclades 2–4, sister clade to subclades 2 (*T. tete*) and 3 + 4 (*T. pradensis* + *T. landinga*, respectively). However, we did not recover the phylogenetic relationship of the clade *brasiliensis* sensu [22] or subclade 5.

The clade 2 of *Trichomycterus* sensu stricto (subclades 10–11) is endemic to the headwaters of the São Francisco, Paraíba do Sul, and Paraná–Paraguay basins, where it is sympatric with the first clade (subclades 1–9). However, based on a combined analysis of molecular and morphological characteristics, it is suggested that the species represents a distinct clade of *Trichomycterus* sensu stricto [22]. Our findings reconfigure the extent of *Trichomycterus* sensu stricto proposed previously [7] to exclude *T. septemradiatus*, *T. pauciradiatus*, *T. reinhardti*, and *T. itatyae*. Recently, it was proposed that these species represent an endemic clade [31]. According to these authors, this clade is represented by 10 species (*T. reinhardti*, *T. funebris*, *T. humboldti*, *T. ingaiensis*, *T. pauciradiatus*, *T. piratymbara*, *T. sainthilairei*, *T. septemradiatus*, *T. anaisae*, and *T. luetkeni*), endemic to the Rio São Francisco basin, Rio Doce headwaters, Espinhaço mountain range, and Paraná (Rio Grande) basin between the Mantiqueira and Canastra mountain ranges. Also, it is attributed the high endemism of these species to the uncommon regional relief [31], with small subdrainages crossed by transversal hills due to geological gaps, supporting a complex paleogeographic scenario favoring isolation of taxa in small areas. This evidence may support Clade 2 (subclades 10–11) as a new genus. However, in keeping to a conservative "lumper" perspective to delimit diversity [32], we have chosen not to propose another genus until evidence beyond what has been explored in the present study comes to light.

### 4.1.2. Clades *Cambeva* and *Scleronema*

The genus *Cambeva* was recovered as the monophyletic and sister clade to *Scleronema*, both sisters to *Trichomycterus* sensu stricto. *Cambeva*, *Scleronema*, and *Trichomycterus* are endemic genera to the eastern neotropical region, occurring in sympatry in São Francisco,

Paraíba do Sul, São Francisco, Paraná, and Uruguay basins. This phylogenetic relationship is congruent with previous studies [6,7,22].

### 4.1.3. Clade *Eremophilus*

Subclade 16, represented by *T. sandovali* (Magdalena) and *T.* cf. *knerii* (Orinoco), was recovered as a sister clade to the eastern Brazilian species [(*Cambeva* + *Scleronema*) + (*Trichomycterus* sensu stricto)]. This subclade exhibits a disjunct distribution since it was not phylogenetically close to the nearby lineages. Subclade 16 occurs in sympatry in the Magdalena basin with subclades 23 (*T. striatus* and *T. straminius*) and 35 (*T. transandianus*). However, despite this sympatry, it shows significant genetic distance values and lacks any close phylogenetic relationship to the other taxa in its vicinity. Previous studies had recovered *T. sandovali* and *T.* cf. *knerii* with *Eremophilus mutisii* from the Magdalena basin [6,22]. These results provide an insight into the high diversity and endemism in these regions, mainly in the right arm of the headwaters of the Magdalena River [33].

### 4.1.4. Clade *Hatcheria*

The genus *Hatcheria* Eigenmann 1927 was originally proposed to include *Hatcheria maldonadoi* [3]. *Trichomycterus areolatus* is a more senior name than *B. maldonadoi* and *H. macraei*. We hereby propose the senior name *Hatcheria areolata* n. comb. to encompass the senior name of *T. areolatus*, while designating *Hatcheria macraei* and *Bullockia maldonadoi* as junior synonyms. Consequently, we recognize the Andean clade as the monotypic *Hatcheria areolata*, morphologically distinctive by the presence of a flattened head with the largest width at the opercular patch of odontodes; wide maxilla; dorsally placed eyes; dorsal fin rays 13–15 branched, and a caudal peduncle strongly compressed laterally [34,35]. *Hatcheria* is easily distinguished from both *Trichomycterus* and *Cambeva* by the dorsal position of eyes (vs. lateral or dorsolateral in *Trichomycterus* and *Cambeva*) and the number of branched dorsal fin rays (13–15) vs. 6–7 in *Trichomycterus* and 7–8 in *Cambeva*.

As a result of our analysis, we have identified a monophyletic trans-Andean group of Trichomycterinae, comprising the austral species *Trichomycterus areolatus*, *Bullockia maldonadoi*, and *Hatcheria macraei*. This group should be recognized as a distinct genus, separate from *Trichomycterus*, *Cambeva*, and *Scleronema*. The phylogenetic topology, associated with the geographic distribution of the lineages and genetic distances between clades, supports the reclassification of *T. areolatus* and *B. maldonadoi* under *Hatcheria*. In this context, our study represents an initial step towards clarifying the taxonomic status of the mentioned species. Based on phylogenetic and molecular evidence, we propose that both *T. areolatus* and *Bullockia maldonadoi* should be reclassified and named as *Hatcheria areolata* n. comb. and *H. maldonadoi* n. comb., respectively.

*Hatcheria macraei* was first described in 1855 in the vicinity of Uspullata, Mendoza, Argentina, on the cis-Andean side, but has also been recorded in the southern Chilean province of Aysén [30]. *Hatcheria areolata* n. comb. is one of the most well-distributed species of Trichomycterinae in austral drainages, and it was described in 1846 in the San Jago River in Santiago and is known to occur in the rivers of the western slopes of central Chile [3]. *Hatcheria maldonadoi* is endemic to trans-Andean drainages in the south-central provinces of Chile [34,35], and it was first described in 1928 in the Nonguen River.

The phylogenetic trees obtained in this study revealed two lineages showing 6% divergence. One lineage includes *H. areolate* + *H. maldonadoi*, while the other comprises the southern Chilean populations of *H. areolate* grouped with *H. macraei*, resulting in *H. areolatus* being polyphyletic. This paraphyly of *H. macraei* and *H. areolate* was also reported in a previous phylogenetic study using *cytb* sequences [30]. Our analysis indicates that the close genetic relationship between these species does not support their classification into separate genera.

An examination of the morphological characters shared by these three Andean species reveals several character states that are not shared with *Trichomycterus* sensu stricto. The cephalic laterosensory system is an uninterrupted supraorbital canal reaching the sensory

pore s1 in the trans-Andean species [36,37] vs. s1 interrupted from s3 in the *Trichomycterus* sensu stricto species of the eastern Brazilian clade. The difference also includes a pectoral fin with absent or rudimentary filament vs. a pectoral fin filament always present in *Trichomycterus* sensu stricto and a flattened caudal peduncle with low depth vs. rounded with high depth in *Trichomycterus* sensu stricto [38].

In a prior study [39], an attempt was made to resolve the trans-Andean species complex by analyzing Trichomycterinae specimens in the British Museum. The study proposed that *Hatcheria* should be considered synonymous with the trans-Andean species of *Trichomycterus*, owing to the challenges in distinguishing between these two groups. Another analysis focused on morphological characters, revealing that the highest retention of plesiomorphic traits within Trichomycteridae occurs among species from trans-Andean Patagonia. This finding implies that the family's ancestry is linked to this particular region [40].

### 4.1.5. Clade *Striatus*

The clade *Striatus* comprises the west major clade and is the sister group to the clade *Hatcheria*. It is endemic to the Magdalena basin, which harbors the highest diversity of lineages of Trichomycteridae in the region.

### 4.1.6. Clade Guiana

The Guiana clade (subclades 24–28) is represented by *T.* cf. *guianense* (clade 24), the Mazaruni region (clade 25), the Potaro region (clade 26), *T. guianense* (clade 27), and *T. conradi* (clade 28), which are endemics and occur in sympatry and nearby the Essequibo ecoregion. We obtained the same topology, confirming the monophyly of the Guiana species, as well as the internal relationships within species and groups, which have been previously reported [6,8,22]. These findings further underscore the region's high diversity and endemism, aligning with the Pleistocene refuges identified for Characiformes [33] and *Trichomycterus* [8].

Other clades are also distributed in the areas surrounding the Guiana ecoregion, mainly in the Magdalena ecoregion, where most of the diversity of the main clades is found. In a previous study [6], a monophyletic group comprising *T. guianense* and other neotropical species from both sides of the Andes in the northern reaches of the Magdalena and Amazon basins was recovered. These included *T. banneaui* Eigenmann, 1912 (Magdalena basin, Colombia); *T. cachiraensis* Ardila-Rodríguez, 2008 (Magdalena basin, Colombia); *T. ruitoquensis* Ardila-Rodriguez, 2007 (Magdalena basin, Colombia); *T. spilosoma* Regan, 1913 (San Juan basin, Colombia); *T. striatus* Meek and Hilbebrand, 1913 (Cana River, Panama); and *T. transandianus* Steindachner, 1915 (Magdalena basin, Colombia). Also, a high species diversity was found in the Pakaraima Mountains of Guyana [8], including new species occurring in sympatry in a restricted region.

### 4.1.7. Clade *Ituglanis*

We recovered six subclades of *Ituglanis* spp. structured geographically in three clades: (1) *I. amazonicus* (Madeira: Amazonas basin) and *I. eichhorniarum* (Paraná–Paraguay basin) more to the west of the sampling distribution; (2) *I. boitata* (Laguna dos Patos basin) and *I. parahybae* (Ribeira do Iguape basin) more to the south of the distribution, and (3) *I. herberti* (Tocantins and Paraná basins) and *I. parkoi* (Tapajós: Amazonas basin) more to the east of the sampling area.

### 4.1.8. Clade *Transandianus*

*Trichomycterus transandianus* (clade I, subclade 35), endemic to the trans-Andean portion of the Magdalena basin, was very distinct from the other Trichomycterinae in the region or any other included in the analysis. Even so, this result provided additional support to the high diversity and endemism of the Magdalena ecoregion, highlighting the area's

importance for the diversification of Trichomycterinae, supporting its identification [33] as an important area of endemism and aquatic diversity.

### 4.1.9. Clade Tapajós

*Trichomycterus* sp2. ([6]; clade J, subclade 36) is endemic to the Tapajós basin, and we did not recover any relationship of this species with the others of the subfamily. However, in previous study [6] this species was related to the clade herein called *Hatcheria* (subclades 17–22), meaning that it may be related to the west major clade.

### 4.2. Efficiency of Cytochrome b

There might be a question regarding whether data from a single locus is sufficient to unveil the evolutionary history of *Trichomycterus*. Despite using only one mitochondrial gene with 999 bp, we successfully confirmed the monophyly of *Trichomycterus* sensu stricto, *Cambeva*, *Scleronema*, and *Ituglanis*, along with their interrelationships. Furthermore, we expanded the scope of our study by utilizing a comprehensive set of georeferenced data, covering a larger sampling area with higher representativeness compared to previous research [6–8,22].

Our data were consistent with previous studies that used five genes ([6], 3.284 bp for 16S, COI, *cytb*, Myh6 and Rag2) and six genes ([7], 4.380 bp for COI, *cytb*, Glyt, Myh6, Rag2 and Sh3px), providing additional support for the high discriminatory potential of *cytb* gene sequences. Several characteristics of *cytb* have enabled the evaluation of recent phylogenetic relationships with robustness. These characteristics include a high number of copies per cell, maternal inheritance, low ancestral polymorphism, and the absence of recombination. Moreover, its evolutionary rates facilitate species-level distinctions. However, caution is advised, as a study of Cichlidae relationships [41] discovered at genetic distances exceeding 15%, saturation levels might pose challenges in the analyses, particularly when assessing the phylogenetic relationships between less closely related genera.

### 4.3. The Magdalena–Sinu Diversity

Our study provides support for the identification of seven additional clades, with seven of them geographically associated with the Amazon region, particularly the Magdalena–Sinu ecoregion. We observed the relationship between two major lineages as follows: (1) Eastern/Magdalena (cis-Andean; subclades 1–16): {[*Trichomycterus* + (*Cambeva* + *Scleronema*)] + subclade 16 (*T. sandovali* and *T. knerii*; *Eremophilus mutisii*)} and (2) Western/Magdalena (cis and trans-Andean; subclades 17–28): [*Hatcheria* + subclade 23 (*T. striatus* and *T. straminius*)] + [Guiana clade (*T. guianense* and *T. conradi*)]. This topology suggests that cladogenesis occurred within the Magdalena ecoregion, as both eastern and western lineages are present in this region. Furthermore, the area exhibits high diversity, serving as Pleistocene refuges to Characiformes [33], which may also explain the high diversity of Trichomycterinae lineages.

### 4.4. The Cladogenesis Processes under a Phylogeographic Perspective

Our analyses brought significant contributions to our understanding of how lineages are organized within a phylogenetic context. Additionally, we established reference values of genetic divergence, aiding in the classification of taxa and serving as a valuable reference for barcoding purposes. Furthermore, we assessed the phylogeographic structure and examined geological evidence to infer processes that have influenced the vicariant pattern. This comprehensive approach enables a better understanding of the current distribution of lineages and sheds light on the ecological and phylogeographic history that underlies the cladogenic processes.

We classified South American species of Trichomycterinae into two geographic regions: (1) Eastern Brazil (clades *Trichomycterus* sensu stricto; *Cambeva*; *Scleronema*; and *T. sandovali* + *T. knerii* + *E. mutisii*); and (2) trans-Andean Austral (clade *Hatcheria*, which includes *H. macraei*, *H. maldonadoi*, and *H. areolata*; *T. striatus* + *T. straminus*; and clade

Guiana, which includes *T. guianense* and *T. conradi*). While prior studies propose similar topologies [6,7], they lack an in-depth discussion of the trans-Andean Austral and Guiana Shield clades. Moreover, these studies do not delve into the phylogeographic perspective regarding the boundaries of the geographic distributions of neotropical lineages.

The Pleistocene period coincides with the great diversification of the main Trichomycterinae lineages. It is plausible that the processes that facilitated the allopatric speciation of Characiformes, with their refugees in the highlands during the glaciations and subsequent colonization of the lowlands, along with interactions between marine incursions, uplift of the paleoarches, and historical connections allowing cross-drainage dispersal, may also account for the speciation processes of Trichomycterinae in the Paraná–Paraguay, Guianas, and Magdalena–Orinoco refuges [33].

The first proposed partition of eastern Brazilian *Trichomycterus* into *Cambeva* and *Trichomycterus* sensu stricto [7] referred to the Andean and northwestern neotropical species as "other Trichomycterinae". Our analyses go further on this topic, discussing the organization of Andean lineages and their biogeographic history.

Based on [42], species richness hotspots include the Chocó region of Colombia, the Pantanal wetlands of the Upper Paraguay basin (central Brazil and Paraguay), and the Atlantic Forest of southeastern Brazil. These regions were also recognized as areas of endemism [33]. Furthermore, our study identifies these areas as significant for comprehending the cladogenesis between *Trichomycterus* sensu stricto and *Cambeva + Scleronema*.

Considering the intrinsic characteristics of these catfish, commonly associated with freshwater headlands [2,3], altitude is a limiting factor for the Trichomycterinae, which may explain their high structuring even in geographically close areas. The main highland areas in the neotropical region, namely the Andes, the Guiana Shield, and the Atlantic coastal drainages, are separated by depressions between the shields [43]. Multiple river systems drain the high plateau of the Guiana Shield in all directions [8]. The Andes, Guiana, and Eastern Brazil regions are situated in the three highest neotropical regions, which were geographically isolated by the Paranaense and Amazon Seas during the Pleistocene glacial maximums [44]. This isolation has significantly influenced the distribution of freshwater fishes, particularly rheophilic fish, which thrive in high-energy waters and are commonly found in the highland regions [45]. In contrast, these fish taxa are typically scarce in lowland plains or floodplains [46], explaining the near absence of species of Trichomycterinae in such environments (except for *Ituglanis*). The different lineages of Trichomycterinae are structured in various neotropical high plains, a pattern that can be attributed to recent events—particularly the formation of the Austral Andean high plains—or to older features like the Guiana and Brazilian Shields [47].

We can posit that these three neotropical highlands function as "lineage islands" (as we termed them) due to the unique life habits of these catfish. The valleys that separate these highlands serve as barriers to dispersal, imposing constraints of gene flow between these "islands". Consequently, understanding vicariant patterns between lineages necessitates a comprehensive examination of geomorphological processes.

Due to significant neotectonic activity in the eastern portion of the Brazilian Shield, the landscape dynamics in the region, particularly in the area of sympatry between *Trichomycterus* sensu stricto and *Cambeva* at the headwaters of the Paraíba do Sul, São Francisco, and Paraná–Paraguay basins, have undergone considerable change. A study focusing on the geomorphological analysis of South America [48] revealed that this region is divided at the 125° Azimuth lineament, which separates the Amazon region from Paraná–Paraguay. The presence of interbasin arches in the landscape, associated with intrusive magmatism, contributed to the formation of divides between the Amazon, São Francisco (draining northward), and Paraná–Paraguay basins (draining southward). These geomorphological events have led to high environmental heterogeneity in the region, which is likely linked to the high genetic diversity observed in the taxa of Trichomycterinae found in the limits of the 125° Azimuth lineament. Moreover, this area served as a refuge for many aquatic

species during the glaciations of the late Miocene for Characiformes, acting as a center of endemism and dispersion [33].

In a previous study [49], the authors presented compelling evidence of a prominent denudational escarpment that separates the São Francisco, Paraná, Doce, and Paraíba do Sul basins, leading to a gradual shift of the lower plateaus towards the interior of the continent. The occurrence of escarpment events renders the headwaters of these rivers more susceptible to denudation, potentially resulting in their isolation or connection over time. Therefore, the diversity within this region may hold greater significance, as it likely facilitates the exchange of diversity events among different portions that were previously isolated.

Also, a study that examined the role of paleodrainages on the distribution pattern of the genetic diversity of *T. alternatus* was conducted [50], and their findings indicated that the intricate distribution patterns of this species were influenced by factors, including paleoriver dispersion, headwater capture, and geographical barriers like the Cabo Frio Magmatic Lineament. These events would explain the complexity and dynamism observed in the Serra do Mar region and the recurrent examples of biogeographic breaks identified in this area, which align with the results we observed in Trichomycterinae.

The South Atlantic Convergence Zone, characterized by extensive cloud cover across the Amazon–Atlantic region, exhibits its most intense influence in the headwaters of the São Francisco, Paraná, Paraíba do Sul, and Doce basins. This can lead to distinct precipitation compared to the other regions and cause climatic anomalies, particularly during austral summer [51]. The interplay of these factors collectively influences the dynamics of the Trichomycterinae diversity in the area.

Neotropical freshwater fish display primary diversity patterns driven by erosion-induced river capture and climate-induced eustatic (global) sea-level oscillations. Landscape evolution processes can alter the spatial dynamics and course of rivers [42], which could account for the disjunct distribution of sister lineages, such as *T. sandovali* + *T.* cf. *knerii* from the Magdalena and Orinoco basins, relative to (*Trichomycterus*) + (*Cambeva* + *Scleronema*) from eastern Brazil.

An interesting finding is the strong support found between the two major Trichomycterinae lineages, with one predominantly distributed in the eastern neotropical region and the second in the western region. However, both lineages also occur in the Magdalena ecoregion, suggesting its significant role in the eastern–western cladogenesis of the subfamily. This observation gains further support when considering the high diversity of lineages in the Magdalena region and neighboring areas (e.g., Guianas), with representation of most of the subfamily's lineages and high species endemism. Furthermore, a distinct pattern emerges, with the western lineages showing connections with the trans-Andean portion, including the Magdalena, while the eastern lineages predominate in the cis-Andean portion, even within that ecoregion. The cladogenesis of these two major lineages was estimated at 28 mya, aligning with the active period of the Andes uplift (40 to 4 mya). The isolation of Orinoco and the Guianas during the Miocene marine incursions [33] likely contributed to the significant difference in fish species from these regions, despite their geographic proximity.

Finally, our findings highlight the remarkable complexity within the Trichomycterinae subfamily. Comprising a highly diverse and widely distributed group, studying their evolutionary history presents unique challenges. The relationships between species of *Trichomycterus* are quite complex and will require the inclusion of the greatest possible representativeness of the genus, particularly considering the characteristics associated with several taxa that were still not yet explored in this study. Nonetheless, our results constitute a significant step towards unraveling monophyletic groupings within the genus. Incorporating additional species previously not included in genetic studies may further refine the clades identified in our investigation. Additionally, an important avenue for further investigation involves identifying species from the central neotropics in relation to

the eastern, western, and northeastern groups, necessitating the inclusion of more species from the genus and the subfamily in phylogenetic analyses.

**Supplementary Materials:** The following supporting information can be downloaded at: https://www.mdpi.com/article/10.3390/d15080929/s1. Figure S1. Phylogenetic tree original of Trichomycterinae based on 566 cytochrome b sequences (405 haplotypes) with 999 bp. Bayesian Inference with values above each branch correspond to the branch consistency index (BPP); Figure S2. Phylogenetic tree original of Trichomycterinae based on 566 cytochrome b sequences (405 haplotypes) with 999 bp. Bayesian Inference with values above each branch correspond to age for clades. as well as the margin of error demonstrated in node bars; Figure S3. Phylogenetic tree original of Trichomycterinae based on 566 cytochrome b sequences (405 haplotypes) with 999 bp. Maximum Likelihood with values above each branch correspond to the branch consistency index (ML bootstrap); Table S1. Type locality of the species nominally allocated to *Trichomycterus* and of the type species of the nine genera of Trichomycterinae. Type species in asterisk; Table S2. Specimens used in the present study, with their respective locations (basin, latitude, and longitude), the collection where specimen is deposited (only for sample exclusive from this study), GenBank accession number of all sequences used on the phylogenetic analysis, and the source that generated the sequences (present study or previous studies). Clade numbers are in according to the phylogenetic tree in Figure 2; Table S3A. Table with percentages of K2P pairwise genetic distances between the subclades of *Trichomycterus* sensu stricto. Values in diagonal (below, dark grey) correspond to the genetic distance between clades by category. Values above (above, blue) refer to the respective standard errors. Bold values on the diagonal refer to the intraclade distance (1 to 31, as referenced in Tables S1 and S3C); References [52–113] are cited in the supplementary materials. Table S3B. Genetic divergence range, by category, and occurrence within subclades, genera, and subfamily; Table S3C. List of alphabetical clade code and respective genus/species and numerical subclade code with respective species.

**Author Contributions:** Conceptualization: T.d.A.V., L.M.S.-S. and V.F.; methodology: T.d.A.V. and V.F.; software: T.d.A.V.; validation: T.d.A.V. and V.F.; formal analysis: T.d.A.V.; investigation: T.d.A.V.; resources: V.F.; data curation: T.d.A.V., M.M., L.M.S.-S. and V.F.; writing-original draft preparation: T.d.A.V., M.M. and V.F.; writing-review and editing: T.d.A.V., M.M., L.M.S.-S. and V.F.; supervision: L.M.S.-S. and V.F.; project administration: V.F.; funding acquisition: V.F. All authors have read and agreed to the published version of the manuscript.

**Funding:** This research was funded by Coordenação de Aperfeiçoamento de Pessoal de Nível Superior (CAPES) for the doctoral bursary granted to T.d.A.V. (1170780/2012-6); Conselho Nacional de Desenvolvimento Científico e Tecnológico (CNPq) for the research bursary granted to L.M.S.S. (PCI-E1 grant 465023/2014-2); and CNPq and Fundação de Amparo à Pesquisa e Inovação do Espírito Santo (FAPES) under Pronem Program to V.F. (CNPq/FAPES 80600417/17).

**Institutional Review Board Statement:** Ethical review and approval were waived for this study due to most of the samples being donated from the museum collections. The Instituto Chico Mendes de Conservação da Biodiversidade (ICMBio) granted collecting license number 41225-1. In this study the collecting and processing specimens as well as access to the genetic heritage was duly registered with SisGen-Sistema Nacional de Gestão do Patrimônio Genético e dos Conhecimentos Tradicionais Associados, as required by Brazilian law.

**Data Availability Statement:** The data underpinning the analysis reported in this paper are deposited at GBIF, the Global Biodiversity Information Facility, and are available at https://ipt.pensoft.net/resource?r=trichomycteridae-ufes-2023 (accessed on 9 May 2023).

**Acknowledgments:** We thank the curators and staff of the collections for providing or helping to acquire specimens and/or tissues used in this study: Luiz Fernando Duboc and Leonardo Ferreira da Silva Ingenito (Coleção Zoológica Norte Capixaba, Universidade Federal do Espírito Santo, CZNC/CEUNES-UFES); Instituto Nacional da Mata Atlântica (INMA); Marcelo Ribeiro de Britto (Museu Nacional do Rio de Janeiro. MNRJ); Daniel Cardoso de Carvalho (Pontifícia Universidade Católica de Minas Gerais, PUC-MG); Carlos Alberto Santos Lucena (Pontifícia Universidade Católica do Rio Grande do Sul, PUC-RS); André Teixeira da Silva (Universidade Estadual do Sudoeste da Bahia, UESB); Jorge Dergam (Universidade Federal de Viçosa, UFV), and Sergio Maia Queiroz Lima (Universidade Federal do Rio Grande do Norte, UFRN). We also thank all the collaborators who

helped with the field activities, Juliana de Freitas Justino for molecular laboratory support, and Victor Vale for helping with statistical analysis.

**Conflicts of Interest:** The authors declare no conflict of interest. The funders had no role in the design of the study, in the collection, analyses or interpretation of data, in the writing of the manuscript, or in the decision to publish the results.

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
