# Peer review of "Pleistocene Aquatic Refuges Support the East–West Separation of the Neotropical Catfish Trichomycterinae (Siluriformes: Trichomycteridae) and High Diversity in the Magdalena, Guiana, and Paraná-Paraguay Basins"

_diversity, doi:10.3390/d15080929_

Round 1

Reviewer 1 Report

Thank you, editors. Please see attachment file.

Author Response

All requested corrections were done, as follow:

1) About Table

Table 1 Caption (pg 6): corrected Table: by Table.

Table 2 Caption (pg 13): Corrected Table 2: by Table 2.

Table 3 Caption (pg 16): Corrected Table 3: by Table 3.

2) About references

The reference formats were revised and are following according to Instructions for Authors.

3) Please confirm definition of using ‘et al’. in references

We listed the complete author names of the references 12, 15, 20 and 115.

4) Reference 18

Added space before Stecher, as requested.

5) order of the references #51 and #52 is reversed.

Order was corrected.

6) Reference #79

Parentheses were deleted, as requested.

7) Please move ‘year’ to after Journal name.

Done for the references #88, as requested

8) Citation of references 93, 113 and 115 are missing

Reference #93 was in duplicity to #113. Then, I maintained the #93 and cited it on Table S2, line #100, Trichomycterus nigricans*.

Reference #115 was originally cited on the Caption of the Figure 3, but is now numbered as reference #113, because the old #114 (Costa, 1992) was also excluded since it was not originally cited in the text.

9) Tables 1 and 2 – use “.” Instead of “,”

Done as requested.

10) Figure 2 Caption: correct Lowercase in the word “Phylogenetic”.

Done as requested.

Reviewer 2 Report

I recommend that this paper be published after editing the grammar.  Scientific names are nouns and cannot be used as adjectives.  I marked this in several places but this needs a global search.  Scientific names should be italicized

Author Response

All requested corrections have been made, and a thorough global search was conducted to edit grammar and scientific name formatting as needed. Please refer to the uploaded revised version for highlighted corrections. Please see also the attachment.
